# Vaccine-Induced Humoral and Cellular Response to SARS-CoV-2 in Multiple Sclerosis Patients on Ocrelizumab

**DOI:** 10.3390/vaccines13050488

**Published:** 2025-04-30

**Authors:** Jelena Drulovic, Olivera Tamas, Neda Nikolovski, Nikola Momcilovic, Vanja Radisic, Marko Andabaka, Bojan Jevtic, Goran Stegnjaic, Milica Lazarevic, Nikola Veselinovic, Maja Budimkic, Sarlota Mesaros, Djordje Miljkovic, Tatjana Pekmezovic

**Affiliations:** 1Clinic of Neurology, University Clinical Center of Serbia, 11158 Belgrade, Serbia; stojiljkovic.olivera@gmail.com (O.T.); nikolamom1993@gmail.com (N.M.); radisic.vanja.95@gmail.com (V.R.); marko_med@yahoo.com (M.A.); n.veselinovich@gmail.com (N.V.); budim17@gmail.com (M.B.); sharlotam@gmail.com (S.M.); 2Faculty of Medicine, University of Belgrade, 11000 Belgrade, Serbia; 3Institute for Biological Research “Sinisa Stankovic”, University of Belgrade, 11000 Belgrade, Serbia; neda.djedovic@ibiss.bg.ac.rs (N.N.); bojan.jevtic@ibiss.bg.ac.rs (B.J.); goran.stegnjaic@ibiss.bg.ac.rs (G.S.); milica.lazarevic@ibiss.bg.ac.rs (M.L.); djordjem@ibiss.bg.ac.rs (D.M.); 4Institute of Epidemiology, Faculty of Medicine, University of Belgrade, 11000 Belgrade, Serbia; pekmezovic@sezampro.rs

**Keywords:** multiple sclerosis, ocrelizumab, COVID-19, vaccine response, seroconversion

## Abstract

**Background/Objectives**: The aim of our study was to investigate B cell and T cell responses in people with multiple sclerosis (PwMS) treated with ocrelizumab, a humanized anti-CD20 antibody, who were vaccinated with second and/or booster doses of various vaccine brands against COVID-19. Additionally, we detected the outcomes related to COVID-19 in PwMS after vaccination, based on follow-up for at least 12 months. **Methods**: We enrolled 91 PwMS on ocrelizumab and 42 healthy controls (HCs) in a prospective, single-center study, conducted at the Clinic of Neurology, UCCS, between January 2022 and October 2024. The serological responses were measured using the spike receptor-binding domain (RBD) Architect SARS-CoV-2 IgG Quant kit (Abbot), and cellular responses were measured by quantifying IFN-γ secretion in blood incubated with SARS-CoV-2 antigens. **Results**: A total of 58.2% (53/91) of PwMS on ocrelizumab and 100% of the HCs (42/42) were seropositive after a second or booster vaccination (*p* < 0.001), irrespective of the vaccine brand received. Anti-spike antibody levels were significantly lower in PwMS on ocrelizumab compared to the HCs (*p* < 0.001), again irrespective of the vaccine type. Interferon-γ responses were detected in 95.6% of the PwMS receiving ocrelizumab therapy and 97.6% of HCs after vaccination (*p* = 0.570). In our cohort, PCR-confirmed SARS-CoV-2 infections after vaccination occurred in a similar proportion of the PwMS (45/91, 49.5%) and HCs (15/32, 46.9%) (*p* = 0.139). Most of the PwMS (36/45, 79.2%) and HCs (13/15, 87.8%) had COVID-19 of mild severity. **Conclusions**: PwMS treated with ocrelizumab developed diminished humoral and robust cellular responses following two and three SARS-CoV-2 vaccinations. The obtained immunity after SARS-CoV-2 vaccination may translate into lower incidence and severity of COVID-19.

## 1. Introduction

Coronavirus disease (COVID-19) is a respiratory illness caused by the SARS-CoV-2 virus [1]. Immunocompromised patients are more likely to develop serious illness and require medical attention. Therefore, since people with multiple sclerosis (PwMS) receive various disease-modifying therapies (DMTs) to prevent disease progression, clinicians were concerned about their possible deleterious effects when combined with COVID-19 infection. Additionally, one of the key questions was whether PwMS receiving DMTs could have an adequate immune response to infection and vaccines. PwMS treated with B-cell depleting anti-CD (aCD20) therapies are prone to infections [2,3,4,5,6]. Therefore, all PwMS are recommended by the MS International Federation to be vaccinated against SARS-CoV-2 [7]. SARS-CoV-2 vaccination induces immune protection against COVID-19 in healthy people, producing both humoral and cellular immune responses [8]. Until now, it has been demonstrated that PwMS treated with certain DMTs, especially fingolimod or B-cell depleting therapies, had diminished humoral and partially cellular responses to the vaccination [9,10,11].

The aim of our study was to investigate the effect of ocrelizumab on B cell and T cell responses in the PwMS treated with ocrelizumab who were vaccinated with second and/or third doses of various vaccines against COVID-19 [mRNA (Pfizer BioNTech, New York, NY, USA) vaccine, vector vaccines (AstraZeneca, Sputnik V, Cambridge, UK) or inactivated BBIBP vaccine (Sinopharm, Beijing, China)]. Additionally, we detected the outcomes related to the prevention of the development of COVID-19 in PwMS patients after vaccination based on the follow-up analyses of the study participants, which lasted for at least 12 months after the last dose of the vaccine.

## 2. Materials and Methods

This investigator-initiated study, sponsored by the Faculty of Medicine University of Belgrade and supported by a financial grant from Roche d.o.o. Beograd, is a prospective, single-center observational cohort study that was conducted in the MS Center at the Clinic of Neurology, University Clinical Center of Serbia, between January 2022 and October 2024. This study was approved by the Ethics Committee of the Faculty of Medicine University of Belgrade (No. 1322/IX-65, 30 September 2021) and was performed according to the Declaration of Helsinki. All participants signed informed consent.

The study included 91 PwMS treated with ocrelizumab and 42 sex- and age-matched healthy controls (HCs) who were vaccinated against SARS-CoV-2 with any of the approved vaccines in Serbia [mRNA, (Pfizer BioNTech, Hong Kong) vaccine, vector vaccines (AstraZeneca, Sputnik V) and inactivated BBIBP viral vaccine (Sinopharm)]. None of the participants had COVID-19 infection before their enrollment in the study. All MS patients enrolled in the study had confirmed diagnoses of relapsing-remitting or primary progressive MS according to the 2017 McDonald criteria [12] and had started treatment with ocrelizumab. Demographic and clinical data were recorded at the time of recruitment.

Since one of the study’s aims was to describe the outcomes related to the prevention of the development of COVID-19 in MS patients and HCs after vaccination with second and/or third doses of various vaccines against COVID-19—categorized as a proportion of patients who developed COVID-19 after a first, second, or third dose of vaccination—participants were followed for at least 12 months after their last dose of the vaccine. A total of 39 PwMS treated with ocrelizumab and 12 HCs had received two doses (primary series) of vaccine, and 52 PwMS and 30 HCs had received a third dose/booster. The Omicron variant, first identified in late 2021, became the dominant strain worldwide in 2022. A study by Novkovic et al., covering the period from March 2020 to the end of January 2023 in the Republic of Serbia, identified the presence of a few variants: Alpha (5.6%), Delta (7.4%), and Omicron (70.3%), with one variant of interest, Omicron recombinant “Kraken” (XBB.1.5) [13]. In the period of late 2023 through 2024, no changes were registered in the dominant strains of SARS-CoV-2 in Serbia, based on information obtained from referent laboratories (personal communication).

The serological and cellular responses to SARS-CoV-2 vaccine were measured at 19.2 ± 2.5 (range 13–23) months and 17.4 ± 2.5 (range 15–23) months, respectively, after the second dose of the vaccine, and at 14.2 ± 3.1 (range 6–23) months and 16.0 ± 2.4 (range 7–20) months, respectively, after the booster. The time between vaccination in PwMS with second doses of the vaccine and blood collection was 19.3 ± 2.2 (range 14–23) months for mRNA vaccines and 18.9 ± 2.7 (range 13–23) months for inactivated viral vaccines (*p* = 0.265); the time between vaccination with a booster and blood collection was 13.5 ± 2.9 (range 6–18) months and 15.2 ± 3.3 (range 10–23) months (*p* = 0.194) for mRNA and inactivated viral vaccines, respectively. Only one person with MS was vaccinated with two doses of a vector vaccine with a time interval between the second dose and blood sampling of 23 months; two PwMS were vaccinated with a booster vector vaccine after 14.5 ± 0.7 months.

All study participants were interviewed by phone by the doctors who participated in the study in order to obtain information about whether they had experienced a COVID-19 infection after vaccination by the end of the study. In the case of a positive answer, adequate medical documentation was submitted, and then the severity of the infection was assessed according to the WHO Working Group on the Clinical Characterization and Management of COVID-19 infection [14].

### 2.1. Detection of SARS-Cov-2 IgG

Peripheral blood samples were collected from PwMS and HCs in lithium heparin (Li-hep) tubes (Neomedica, Nis, Serbia). Serum was obtained using centrifugation at 1000× *g* for 20 min at room temperature. Sera were stored at −20 °C until the detection of IgG specific for S1/RBD SARS-CoV-2 S protein. Serum levels of the abovementioned IgG were determined as AU/mL using the SARS-CoV-2 IgG Quant kit (Abbot, Waukegan, IL, USA). Measurements based on chemiluminescence microparticle immune assay (CMIA) were performed using an Architect i2000sr Plus immunoassay analyzer (Abbot, Abbott Park, IL, USA) following the manufacturer’s instructions. Values below 50 AU/mL were considered negative, and the upper limit of quantification was 40,000 AU/mL.

In vitro restimulation of antigen-specific peripheral blood T cells with SARS-CoV-2 peptide pools was carried out.

Antigen-specific T cell responses were investigated using PepTivator^®^ SARS-CoV-2 Select—Premium Grade peptide pools (Miltenyi Biotec, Cologne Germany). This pool consists of 88 lyophilized peptides ranging from 9 to 22 amino acids in length, comprising both MHC class I (*n* = 63) and class II (*n* = 25) epitopes. The peptides were derived from the structural (S, M, N, E) and non-structural proteins of SARS-CoV-2. Peripheral blood samples were collected from the PwMS and HCs in lithium heparin (Li-hep) tubes (Neomedica). A total of 600 µl of whole blood was added to 24-well plates (Sarstedt, Germany), either in the absence (negative control) or in the presence of 1 µg/mL PepTivator. The samples were incubated for 24 h at 37 °C in a humidified incubator with 5% CO2. Afterwards, they were centrifuged, and the collected plasma samples were frozen and stored at −80 °C.

### 2.2. ELISA

IFN-γ concentration in plasma samples was determined by the sandwich ELISA method using MaxiSorp plates (Nunc, Roskilde, Denmark). Anti-human IFN gamma Monoclonal Antibody (2G1) and anti-human IFN gamma Monoclonal Antibody (B133.5) biotin (Thermo System Scientific, Waltham, MA, USA) were used for the detection of human IFN-γ, following the manufacturer’s instructions two times for each specimen. Values above 30 pg/mL were considered positive.

### 2.3. Statistical Methods

In this investigation, an open cohort was used because the inclusion criteria are vaccination against SARS-CoV-2 and/or COVID-19 infection.

Continuous variables, depending on the type of distribution, are reported as mean ± standard deviation (SD) or median with interquartile ranges (IQR). Correlation analysis between variables related to humoral and cellular immunity after the vaccination is performed using non-parametric correlation coefficients.

## 3. Results

### 3.1. Demographic and Clinical Characteristics of the Participants

We recruited for this prospective study 91 PwMS treated with ocrelizumab and 42 HCs who were vaccinated with various vaccines against SARS-CoV-2 with two or three doses. Table 1 summarizes their demographic and clinical characteristics.

There was no significant difference between PwMS and HCs regarding gender, age, vaccine brand, and number of vaccine doses.

### 3.2. Serological Response in Vaccinated Participants

Table 2 summarizes anti-spike antibody titers and SARS-CoV-2 antibody responses for each vaccine in PwMS on ocrelizumab and HCs. Data from 91 PwMS treated with ocrelizumab and 42 HCs were included. Antibody responses to SARS-CoV-2 were detected in 100% of HCs (42/42) and 58.2% (53/91) of PwMS on ocrelizumab (*p* < 0.001), irrespective of the vaccine brand received. For HCs, vaccination with all vaccine brands resulted in a 100% serological response, with positive antibodies (42/42). Anti-spike antibody levels were significantly lower in PwMS treated with ocrelizumab compared to the HCs (*p* < 0.001), irrespective of the vaccine type. Two PwMS who received vector vaccine had the highest anti-spike antibody titer followed by those who received mRNA vaccine, while the lowest titer was detected in those who received inactivated vaccine. On the contrary, PwMS on ocrelizumab who received the inactivated viral vaccine had the highest rate of COVID-19 seropositivity at 67.6% (25/37), with 52.9% (27/51) for mRNA vaccines and 1/2 for the vector vaccines (*p* > 0.05). A statistically significant difference in the titers between mRNA and inactivated vaccines was not found (*p* > 0.05).

Gender and age were not associated with either the anti-spike antibody titer or the positive response, in either PwMS or the HCs.

### 3.3. SARS-CoV-2 Specific T-Cellular Immune Response After Vaccination

In ocrelizumab-treated PwMS, interferon-γ release—as a parameter of the cellular immune response—was decreased after vaccination compared to the HCs, but the differences between the groups were not significant (*p* = 0.591). The percentage of the PwMS on ocrelizumab who presented a cellular response after vaccination above the predefined cut-off level was 95.6%. Similarly, 97.6% of HCs also had a positive cellular response after vaccination (*p* = 0.570).

Statistically significant differences were found neither in the level of interferon-γ release nor in the proportion of a positive cellular response after vaccination between mRNA and inactivated vaccines (*p* > 0.05) (Table 3 and Figure 1).

Out of 91 ocrelizumab-treated PwMS, 38 did not develop SARS-CoV-2 specific antibodies. In this subgroup of PwMS without a seropositive response, 35 had a positive cellular response after vaccination. Thus, only 3 patients had neither serological nor cellular immune responses after vaccination (two with the Sinopharm vaccine and one with the Pfizer vaccine). On the other hand, all HCs developed SARS-CoV-2-specific antibodies, and 97.6% developed cellular immune responses after vaccination. In PwMS, there is a weak positive non-significant correlation between the development of serological and cellular immune response (ρ = 0.145, *p* = 0.172).

### 3.4. SARS-CoV-2 Infections in Patients with Ocrelizumab and Healthy Controls

In our cohort, PCR-confirmed SARS-CoV-2 infections after vaccination occurred in a similar proportion between the PwMS (45/91, 49.5%) and HCs (15/32, 46.9%) (*p* = 0.139). Eight PwMS and two HCs were infected twice with SARS-CoV-2. Analysis of the COVID-19 infection severity showed that most of the PwMS (79.2%) and HCs (87.8%) had an infection of mild severity, and 20.7% of the PwMS and 12.1% of the HCs (*p* = 0.321) had a moderate severity infection. Eight PwMS (8.8%) and two HCs (4.8%) were hospitalized for COVID-19 during the study. In four PwMS, high-flow nasal oxygen therapy was applied, and four received COVID-19-specific antivirals. No study participant was intubated, admitted to the intensive care unit, or died.

The duration between booster vaccination and blood sampling had no effect on cellular response (*p* = 0.965). In the total cohort, the mean time period between booster vaccination and blood sampling was 16.3 ± 3.3 months.

Our analysis of the COVID-19 infection severity, defined according to the WHO Working Group on the Clinical Characterization and Management of COVID-19 infection (2020), showed that most of the PwMS (36/45, 79.2%) and HCs (13/15, 87.8%) had an infection of mild severity. Moderate severity of infection was registered in 9/45 PwMS and 2/15 control subjects (*p* = 0.847).

## 4. Discussion

In our investigation, we studied the regulation of humoral and cellular immune response by mRNA, inactivated, and vector vaccines in 91 PwMS undergoing treatment with ocrelizumab and 42 healthy vaccine recipients in order to assess the magnitude of vaccine-induced protective immune responses and the clinical efficacy of vaccination. Our study adds a few novelties to our knowledge. To the best of our knowledge, this is the first study comprising an equal proportion of anti-CD20 treated PwMS and HCs vaccinated with mRNA and inactivated virus vaccines against COVID-19. Additionally, it provides data related to the maintenance of robust cellular response after a long period of time period post-vaccination—up to 20 months—in PwMS treated with ocrelizumab and HCs, with humoral response being persistently diminished in ocrelizumab treated patients compared to HCs. Finally, it offers evidence related to the clinical efficacy of vaccination against COVID-19 infection, showing that the number of SARS-CoV-2 infections after vaccination was similar in PwMS on ocrelizumab in comparison to HCs (*p* = 0.139).

In our study, in serum samples collected during the time period ranging from 6 to 23 months (mean 14 months) after vaccination with two or three doses of vaccines, we found that antibody responses to SARS-CoV-2 were detected in 100% of the HCs and 58.2% of the PwMS on ocrelizumab (*p* < 0.001), irrespective of the vaccine brand received. As already demonstrated by other authors, our study shows decreased anti-spike IgG antibody titers in PwMS on ocrelizumab in comparison to HCs (*p* < 0.001) [15,16,17,18,19,20,21]. PwMS who received an mRNA vaccine had higher anti-spike titers compared to those who received an inactivated vaccine, with the difference not reaching statistical significance. On the other hand, PwMS on ocrelizumab who received the inactivated vaccine had a higher rate of COVID-19 seroconversion at 67.6%, compared to 52.9% in mRNA vaccines, again not reaching statistical significance. Thus, similar to the immunogenicity data presented in previous studies in the populations receiving mRNA or inactivated vaccines [17,18,19], we confirmed that the humoral response to both inactivated and mRNA vaccination observed in PwMS on ocrelizumab is blunted [20].

Analysis of cellular responses after vaccination in our study showed that 95.6% of PwMS on ocrelizumab therapy and 97.6% of HCs presented a protective response above the predefined cut-off after vaccination (*p* = 0.570). There was only a trend observed for a slightly decreased level of interferon-γ release in PwMS treated with ocrelizumab compared to HCs, and the difference was not significant (*p* = 0.591). Findings from our investigation demonstrated that the majority of PwMS show a significant T cell response to SARS-CoV-2 vaccination. To date, it has already been shown that PwMS on anti-CD20 are mounting an even more robust T cell response than those who were not treated with DMTs or were treated with other DMTs [11,22,23]. Additionally, the positive impact of a third dose on cellular immunity in PwMS on anti-CD20 has been reported in most prior studies [21,24,25,26]. Finally, it has to be emphasized that our findings support recently published data that the cellular response to SARS-CoV-2 proteins remained elevated up to 12–14 months after the booster [27].

Very recently, researchers have investigated humoral and cellular responses in PwMS on ocrelizumab before and after the first to fourth mRNA SARS-CoV-2 vaccinations and its relationship with breakthrough infection [28]. This longitudinal study demonstrated low levels or the absence of anti-RBD following vaccination, with a remarkable increase observed after infections and the fourth dose of vaccine. Within the same period, a subset of PwMS (75%) suffered from infections. The T-cell response remained robust throughout the study.

It has been demonstrated that various DMTs have a different impact on generating a vaccination response to SARS-CoV-2, with certain therapies predominantly affecting the humoral response, such as anti-CD20, i.e., ocrelizumab, and others affecting cellular immunity [29]. On the other hand, it has previously been clearly shown that, for example, interferon beta does not impair either the humoral or the cellular response to SARS-CoV-2 vaccination [30]. As already mentioned, in line with our results, a reduced humoral response has been identified in ocrelizumab-treated PwMS, along with a robust T cell response [29]. Our results confirm earlier observations [10,23,29,31] in a large cohort, including the existence of robust cellular immune response both in PwMS treated with ocrelizumab and HCs. However, it is uncertain how this data translates to the clinical setting. To date, only a few studies have investigated whether a blunted humoral response and a strong cellular immune response after vaccination may potentially prevent severe COVID-19 infection [27,32,33].

Thus, Kister et al. investigated the impact of SARS-CoV-2-specific immunity in vaccinated, ocrelizumab-treated MS patients on the development and severity of COVID-19 infection [27]. In this longitudinal study, it was shown that in sixty ocrelizumab treated MS patients, the booster vaccine induced a 4- to 5-fold increase in cellular response, which was maintained for up to one year [27]. Sixty percent of the PwMS in the study experienced COVID-19 infection after a second and third dose, and all had infections of relatively mild severity. None of the patients had respiratory failure, and only 8% were hospitalized. In line with these results, in our cohort, PCR-confirmed SARS-CoV-2 infections after vaccination occurred in 45/91 PwMS (49.5%), in comparison with 15/32 HCs (46.9%) (*p* = 0.139). Our analysis of COVID-19 infection severity showed that most of the PwMS (79.2%) had an infection of mild severity and that 20.8% had a moderate severity infection. Eight PwMS (8.8%) and two HCs (4.8%) were hospitalized for COVID-19 during the study. The relative mildness of COVID-19 in ocrelizumab-treated patients in both cohorts might be at least partially due to the protective effect of vaccination and the availability of effective anti-SARS-CoV-2 therapies. Similarly, Novak et al. demonstrated that 59% of PwMS on ocrelizumab experienced breakthrough infections after the third SARS-CoV-2 mRNA vaccine without developing severe disease courses [32]. In all infected participants, the infection was mild. Finally, Aiello et al. investigated the incidence and severity of COVID-19 breakthrough infections in 64 PwMS on various DMTs after a booster dose of mRNA vaccine [33]. Of this cohort, 17 participants were treated with ocrelizumab, and breakthrough infections occurred more frequently among them (11/17, 64.7%) compared to those under fingolimod, interferon-beta, and cladribine. A total of 39.3% of the infections in this cohort were described in participants treated with ocrelizumab, as opposed to 35.7% with fingolimod. The majority of infections were mild (75%) and 17.9% were moderate; only two cases required hospitalization (one on ocrelizumab).

The main limitation of this study is its cross-sectional design. A longitudinal study with sequential, planned sample collection after each vaccine dose for every subject would be preferable. Therefore, the serum samples taken after both the second and the third vaccines were collected during variable and rather unequal time periods. Thus, the assessment of humoral and cellular responses does not represent the peak of post-vaccination immunity for each study participant. Another limitation is related to using IFN-γ release as a readout method. IFN-γ is a well-established marker of inflammation in response to viral infections, it also plays a key role in other inflammatory and immune-mediated conditions. It has not yet been clarified whether additional cytokines could be measured to better define the T cell response to SARS-CoV-2 vaccines. Recently, a study by Al Rahbani et al. suggested that assessing IL-2 and IL-5 levels alongside IFN-γ could provide a more comprehensive characterization of vaccine-induced cellular immunity [34]. This implies a limitation to our study, as our analysis does not account for these additional cytokines, which may offer a broader understanding of immune responses to vaccination.

## 5. Conclusions

This study provides novel information regarding the translation of humoral and cellular immune responses to various types of COVID-19 vaccines in PwMS on ocrelizumab into clinical practice. In this study of a large cohort of PwMS treated with ocrelizumab and vaccinated with mRNA or inactivated SARS-CoV-2 vaccines, we detected a blunted humoral response but a robust cellular response to SARS-CoV-2. Additionally, our investigation demonstrates that the immunity obtained after SARS-CoV-2 vaccination may translate into lower incidence and severity of COVID-19. Further studies with an adequate longitudinal design are warranted in order to confirm the longevity of the post-vaccination response in anti-CD20 PwMS.

## Figures and Tables

**Figure 1 vaccines-13-00488-f001:**
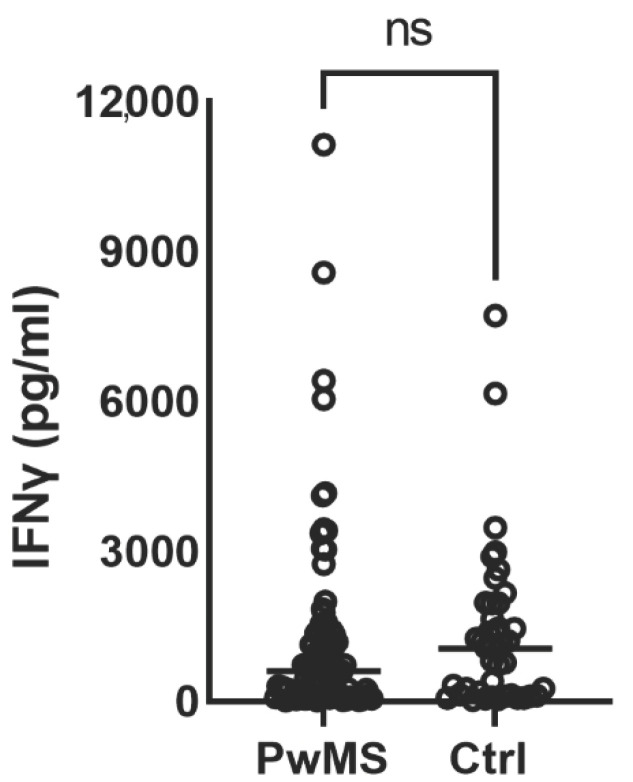
SARS-CoV-2 cellular response. IFN-γ levels to SARS-CoV-2 antigen were measured in plasma obtained from PwMS and HCs using ELISA. Data are presented as mean ±  SD, (*n* = 91 PwMS, *n*= 42 HC), ns—not significant. People with multiple sclerosis—PwMS; Healthy controls—HCs.

**Table 1 vaccines-13-00488-t001:** Demographic and clinical characteristics of study participants.

Variable	PwMS (*n* = 91)	HCs (*n* = 42)	*p*
Gender (% females)	73.6	81.0	0.358
Age (years, mean, SD)	48.4 ± 9.1	47.3 ± 10.5	0.534
Vaccine brand (N, %)			
- Pfizer	51 (56.0)	20 (47.6)	0.700
- Sinopharm	37 (40.7)	21 (50.0)	
- Sputnik V	2 (2.2)	1 (2.4)	
- AstraZeneca	1 (1.1)	0	
Number of vaccine doses (N, %)			
- one	1 (1.1)	0	0.347
- two	38 (41.8)	11 (28.2)	
- three	52 (57.1)	28 (71.8)	

Legend: People with multiple sclerosis—PwMS; Healthy controls—HCs.

**Table 2 vaccines-13-00488-t002:** Anti-spike antibody titers and SARS-CoV-2 antibody responses in the study participants.

	SARS-CoV-2 Antibody Response			
	All	mRNA Vaccine	Inactivated Vaccine	Vector Vaccines
PwMS				
- Anti-spike antibody titer	1216.3 ± 2543.9	1297.6 ± 2174.7	800.8 ± 2048.5	4959.4 ± 8548.6
- % (N) positive	58.2 (53/91)	52.9 (27/51)	67.6 (25/37)	50.0 (1/2)
HCs				
- Anti-spike antibody titer	6104.1 ± 7137.3	8462.5 ± 9178.1	4035.3 ± 3695.0	2381.0
- % (N) positive	100 (42/42)	100 (20/20)	100 (21/21)	100 (1/1)

Legend: People with multiple sclerosis—PwMS; Healthy controls—HCs.

**Table 3 vaccines-13-00488-t003:** Cellular responses to vaccination with different vaccine brand in ocrelizumab-treated patients and healthy controls.

	SARS-CoV-2 Cellular Response			
	All	mRNA Vaccine	Inactivated Vaccine	Vector Vaccines
PwMS				
- Interferon-γ release to SARS-CoV-2 Ag (pg/mL)	1142.1 ± 1820.8	1451.6 ± 2167.0	740.9 ± 1206.5	826.7 ± 144.5
-% (N) positive	94.5 (86/91)	98.0 (50/51)	91.9 (34/37)	100.0 (2/2)
HCs				
- Interferon-γ release to SARS-CoV-2 Ag (pg/mL)	1318.0 ± 1594.5	1065.5 ± 1079.4	1543.1 ± 1995.7	1641.8
- % (N) positive	97.6 (41/42)	100 (20/20)	95.2 (20/21)	100 (1/1)

Legend: People with multiple sclerosis—PwMS; Healthy controls—HCs.

## Data Availability

Data are contained within the article. Additional information is available on request from the corresponding author.

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
