# Peer review of "Vaccine-Induced Humoral and Cellular Response to SARS-CoV-2 in Multiple Sclerosis Patients on Ocrelizumab"

_vaccines, 2025, doi:10.3390/vaccines13050488_

Round 1

Reviewer 1 Report

Comments and Suggestions for Authors

This is an interesting study evaluating the B cell and T cell responses in people with PwMS treated with ocrelizumab after getting second and/or booster doses of COVID-19 vaccines. There are some concerns and suggestions listed below:

  1. In “ Materials and methods” the authors mentioned that “None of the participants had COVID-19 infection before the enrollment in the study.” Do the authors have any diagnosis data to support this statement
  2. In the “ Materials and methods” “data not shown”. The data should be shown
  3. During the sampling period (January 2022 and October 2024), what kinds of SARS-Cov-2 infecting the participants after vaccination need to be addressed.
  4. If the methods or kits used by authors are able to detect different IgG antibodies triggered by different SARS-CoV-2 variants (?) without introducing any bias?
  5. How did the authors perform the ELISA to measure the “interferon-γ”? Are they “specifically resulting from the vaccination nor SARS-CoV-2 infection?
  6. Strongly recommend to plot each value of the anti-spike antibody or interferon-gamma released in graphs.

In addition, it is better to include a line number, which would make it much easier to prepare the reviewer report.

Author Response

This is an interesting study evaluating the B cell and T cell responses in people with PwMS treated with ocrelizumab after getting second and/or booster doses of COVID-19 vaccines. There are some concerns and suggestions listed below:

  1. In “ Materials and methods” the authors mentioned that “None of the participants had COVID-19 infection before the enrollment in the study.” Do the authors have any diagnosis data to support this statement

ANSW: We have no laboratory data to support this statement. It was based on the detailed history taken from each patient regarding the clinical manifestations of the potential  COVID-19 infection.

  1. In the “ Materials and methods” “data not shown”. The data should be shown

ANSW: We added data.

  1. During the sampling period (January 2022 and October 2024), what kinds of SARS-Cov-2 infecting the participants after vaccination need to be addressed.

ANSW: Thank reviewer for the very important comment, we added the answer in the section Material and Methods:

The Omicron variant, first identified in late 2021, became the dominant strain worldwide in 2022. A study by Novkovic et al., covering the period from March 2020 to the end of January 2023 in the Republic of Serbia, identified the presence of few variants: Alpha (5.6%), Delta (7.4%), and Omicron (70.3%), with one variant of interest, Omicron recombinant “Kraken” (XBB.1.5). In the period of late 2023 and 2024, no changes were registered in the dominant strains of SARS-CoV-2 in Serbia, based on information obtained from referent laboratories (personal communication).

Ref: Novkovic M, Banovic Djeri B, Ristivojevic B, Knezevic A, Jankovic M, Tanasic V, Radojicic V, Keckarevic D, Vidanovic D, Tesovic B, Skakic A, Tolinacki M, Moric I, Djordjevic V. Genome sequence diversity of SARS-CoV-2 in Serbia: insights gained from a 3-year pandemic study. Front Microbiol. 2024 Feb 27;15:1332276. doi: 10.3389/fmicb.2024.1332276.

  1. If the methods or kits used by authors are able to detect different IgG antibodies triggered by different SARS-CoV-2 variants (?) without introducing any bias?

ANSW: To the best of our knowledge, kits used cannot detect different IgG antibodies triggered by different SARS-CoV-2 variants.

  1. How did the authors perform the ELISA to measure the “interferon-γ”? Are they “specifically resulting from the vaccination nor SARS-CoV-2 infection?

ANSW: Interferon-gamma release assays (IGRAs) ELISA detect interferon-γ released by T cells in response to the specific viral antigens. Difference between SARS-CoV-2 infection and vaccination depends on the antigen(s) used in the IGRA. With the ELISA method used in this paper, it’s not possible to determine whether the measured IFN-γ is the consequence of the vaccination or SARS-CoV-2 infection, because the peptides from the assay originate from structural proteins (S, M, N, E), as well as non-structural proteins.

  1. Strongly recommend to plot each value of the anti-spike antibody or interferon-gamma released in graphs.

ANSW: Corrected according to the suggestion.

In addition, it is better to include a line number, which would make it much easier to prepare the reviewer report.

ANSW: We put line numbers, however, unfortunately, for some reason we cannot see the line numbers in the text. Please, accept our apologies.

Reviewer 2 Report

Comments and Suggestions for Authors

In the manuscript "Vaccine-induced humoral and cellular response to SARS-CoV-2 in multiple sclerosis patients on ocrelizumab”, Drulovic et al proposed that the obtained immunity after SARS-CoV-2 vaccination may translate into lower incidence and severity of COVID-19, based on that the people with multiple sclerosis (PwMS) treated with ocrelizumab developed diminish humoral and robust cellular responses following two and three SARS-CoV-2 vaccinations. This very short report through determining IFN-γ concentration in plasma samples and detecting SARS-CoV-2 IgG established information regarding translation of humoral and cellular immune response to various types of COVID-19 vaccines in PwMS on ocrelizumab into the clinical practice. Personally, this reviewer thinks that the results of the analysis are too few to understand more. This study did not update recent works, with only one reference published in 2024. Given many reports published recently, the authors should compare and discuss them with this study. The adverse effect following SARS-CoV-2 vaccinations (%) may need to report if any for correlation. The similarity index should be lower. Is any evidence support cellular responses to vaccination did not differ by age, gender, EDSS or type of vaccine (data not shown). Overall, this short report is worth of publication in vaccines.

2.1. Detection of SARS-Cov-2 IgG; should be CoV.

Author Response

In the manuscript "Vaccine-induced humoral and cellular response to SARS-CoV-2 in multiple sclerosis patients on ocrelizumab”, Drulovic et al proposed that the obtained immunity after SARS-CoV-2 vaccination may translate into lower incidence and severity of COVID-19, based on that the people with multiple sclerosis (PwMS) treated with ocrelizumab developed diminish humoral and robust cellular responses following two and three SARS-CoV-2 vaccinations. This very short report through determining IFN-γ concentration in plasma samples and detecting SARS-CoV-2 IgG established information regarding translation of humoral and cellular immune response to various types of COVID-19 vaccines in PwMS on ocrelizumab into the clinical practice. Personally, this reviewer thinks that the results of the analysis are too few to understand more. This study did not update recent works, with only one reference published in 2024. Given many reports published recently, the authors should compare and discuss them with this study. The adverse effect following SARS-CoV-2 vaccinations (%) may need to report if any for correlation. The similarity index should be lower. Is any evidence support cellular responses to vaccination did not differ by age, gender, EDSS or type of vaccine (data not shown). Overall, this short report is worth of publication in vaccines.

ANSW: Thank reviewer for the very important suggestions, we added the sentences in the section Discussion, based on the recent publications.

The study participants did not report any severe adverse effect following SARS-CoV-2 vaccinations. We did not follow mild post vaccination adverse effects, since it was not the focus of our investigation.

In order to reduce similarity index, we made significant changes throughout the text.

We excluded the sentence ‘’Cellular responses to vaccination did not differ by age, gender, EDSS or type of vaccine (data not shown)’’ from the text (section Results) because only 4 PwMS did not develop cellular response and therefore a statistical test would not be precise.

2.1. Detection of SARS-Cov-2 IgG; should be CoV.

ANSW: Corrected.

Reviewer 3 Report

Comments and Suggestions for Authors

Overall this is an excellent paper. It is critically important to determine how antibody therapies affect vaccine efficacy. The data are clearly presented and conclusions justified. The experimental procedures are well described and the data straight forwardly presented. There could be a little more discussion of IFNgamma expression as a marker of T-cell immunity. While it is indeed a reasonable marker of immune recognition, if one has to choose a single cytokine, it does not actually reflect a T-cell function (such as killing of infected cells) and can be associated with inflammation, so that should be stated as a limitation of the data. The manuscript would also be easier to understand by explaining that ocrelizumab is an anti-CD20 antibody in the abstract. It can be assumed in the introduction, but is not clearly stated.

Some editing would also be helpful since there are some wordy and awkward sentences for instance:

Page 7 in the text:

In this study, we have demonstrated in the large cohort of PwMS treated with ocrelizumab vaccinated with mRNA and inactivated vac-cines, consistently blunted humoral response and robust cellular response to SARS-CoV-2 vaccination in majority of PwMS

Could be re-written:

In this study of a large cohort of PwMS treated with ocrelizumab and vaccinated with mRNA or inactivated SARS-CoV-2, we detected a blunted humoral response, but robust cellular responses to SARS-CoV-2.

Also, page 6 line 5 is missing an “a”.

Author Response

Overall this is an excellent paper. It is critically important to determine how antibody therapies affect vaccine efficacy. The data are clearly presented and conclusions justified. The experimental procedures are well described and the data straight forwardly presented. There could be a little more discussion of IFNgamma expression as a marker of T-cell immunity. While it is indeed a reasonable marker of immune recognition, if one has to choose a single cytokine, it does not actually reflect a T-cell function (such as killing of infected cells) and can be associated with inflammation, so that should be stated as a limitation of the data.

ANSW: Thank reviewer for the very important comment, we added the answer in the section Discussion:

Another limitation regards to IFN-γ release, as a readout method. IFN-γ is a well-established marker of inflammation, not only in response to the viral infections, since this molecule also plays a key role in other inflammatory and immune-mediated conditions. It has not been clarified yet whether additional cytokines could be measured to better define the T cell response to SARS-CoV-2 vaccines. Recently, the study by Al Rahbani et al. suggests that assessing IL-2 and IL-5 levels alongside IFN-γ could provide a more comprehensive characterization of vaccine-induced cellular immunity [32]. This implies a limitation in our study, as our analysis does not account for these additional cytokines, which may offer a broader understanding of immune responses to vaccination.

The manuscript would also be easier to understand by explaining that ocrelizumab is an anti-CD20 antibody in the abstract. It can be assumed in the introduction, but is not clearly stated.

ANSW: We added explanation in the Abstract.

Some editing would also be helpful since there are some wordy and awkward sentences for instance:

Page 7 in the text:

In this study, we have demonstrated in the large cohort of PwMS treated with ocrelizumab vaccinated with mRNA and inactivated vac-cines, consistently blunted humoral response and robust cellular response to SARS-CoV-2 vaccination in majority of PwMS

Could be re-written:

In this study of a large cohort of PwMS treated with ocrelizumab and vaccinated with mRNA or inactivated SARS-CoV-2, we detected a blunted humoral response, but robust cellular responses to SARS-CoV-2.

ANSW: Corrected.

Also, page 6 line 5 is missing an “a”.

ANSW: Corrected.

Round 2

Reviewer 1 Report

Comments and Suggestions for Authors

no more comments